

# Influence of Geographic Coordinate System on Weather Simulations of Atmospheric Emissions

Yanni Cao[1,2,4], Guido Cervone[1,2,4,5], Zachary Barkely[3], Thomas Lauvaux[3], Aijun Deng[3], and Alan Taylor[2]

[1]Goeinformatics and Earth Observation Laboratory, The Pennsylvania State University, University Park, PA
[2]Department of Geography, The Pennsylvania State University, University Park, PA
[3]Department of Meteorology, The Pennsylvania State University, University Park, PA
[4]Institute for CyberScience, The Pennsylvania State University, University Park, PA
[5]Research Application Laboratory, National Center for Atmospheric Research, Boulder CO

*Correspondence to:* Guido Cervone (cervone@psu.edu)

**Abstract.** Numerical atmospheric models can generate simulations at very high spatial and temporal resolutions. Many of such models, including the Weather Research and Forecasting (WRF), assume a spherical geographic coordinate system to represent the data and for their computations. However, most if not all Geographic Information System (GIS) data use a spheroid datum because it best represents the surface of the earth. WRF and other numerical systems, simply assume that GIS layers can be input as if they were in a spherical coordinate system.

The difference when reprojecting standard GIS layers onto a sphere can result in latitudinal errors of up to 21 km in the mid-latitudes. Recent studies have suggested that for very high resolution applications, there might be an impact when the GIS input data (e.g. terrain land use, orography) are not reprojected. These impacts introduced by the difference in coordinate systems remain unclear. This research investigates the role and importance of reprojecting GIS layers used by WRF as input by performing sensitivity studies of greenhouse gas transport and dispersion in Northeast Pennsylvania.

## 1 Introduction

Geographic Information Science (GISc) datasets are usually projected on a spheroid coordinate system such as World Geodetic System 1984 (WGS84) or North American Datum 1983 (NAD83). True earth is an irregular spheroid, and these datums are used to approximate the true oblate spheroid geometry of the earth, which flattens the poles and bulges at the equator. Those spheres and spheroids are used in combination with different projections (e.g. Universal Transverse Mercator (UTM), Lat-Lon, Albert Equal Area) to map a 3D view of the earth onto a 2D plane.

Atmospheric models are based on a spherical coordinate system because it usually leads to faster computations and an easier representation of data (Monaghan et al., 2013). This different geographic coordinate systems (GCS) can affect the model results due to the difference in representation of the earth's shape. This difference can lead to latitudinal shifts up to 21 km in the mid latitude (Monaghan et al., 2013). This paper addresses the use of GCS which use a more correct representation of the earth when compared to a simple sphere, as usually assumed by atmospheric models.



In a GCS earth is represented as either an oblate spheroid or a sphere whereas in a spherical system, the earth is always represented as a sphere (Bugayevskiy and Snyder, 1995). This means that when using a spherical coordinate system, the spatial relationships between points on the surface of the earth are altered. The shift in spatial relationship results in a latitudinal error and is consistent across all data that are commonly used as input layers in the atmospheric models such as the Weather Research
and Forecasting (WRF) model. Consequently, numerical errors are introduced by computations carried out in WRF that are a function of latitude such as the Coriolis Force and the incoming solar radiation. As already explained in the Monaghan et al. (2013), a minor mismatch between the WRF model global atmosphere input and static variables will affect the simulation result. Figure 1 shows the latitudinal errors introduced when representing a point on the surface of the earth with a spherical GCS. Point A represents data projected on a spheroid system (red line). When that same point A is represented on a sphere
(green line) like in an atmospherical model, its location gets incorrectly shifted to the point B. The point C is the true location of the point A when correctly projected in the spherical coordinate system. Figure 2 shows that the errors between spheroid and sphere representation for the same point is a function of latitude. The maximum errors occurs at mid latitude, precisely at $45\,°$ N and S.

Differences in coordinate systems and the resulting spatial errors, such as the example provided in Figure 1, have not been a
primary focus in atmospheric modeling because of the relative coarse spatial resolution of the simulation domains (David et al., 2009). More recently, due to the improvements in computational resources and technological advances, atmospheric models are routinely run at higher spatial resolution. Yet this trend in running simulation with high resolution input datasets do not take into account the shift between the coordinate systems which may cause spatial errors in the model's output.

Monaghan et al. (2013) investigated errors caused by different coordinate systems using WRF run with higher resolution
topography and land use datasets over Colorado. Multiple WRF simulations were performed to study differences in meteorological parameters such as air temperature, specific humidity and wind speed. They concluded that the GCS transformation from WGS84 GCS to a spherical earth model caused the input data to shift up to 20 km southward in central Colorado. The impact of this shift leads to significant localized effects on the simulation results. The root mean square difference (RMSD) for air temperature is 0.99 °C, for specific humidity is 0.72 g kg$^{-1}$ , for wind speed is 1.20 m s$^{-1}$. It was
concluded that for high resolution atmospheric simulations, the issue resulting from datum and projection errors is increasingly important to solve. All datasets used as input should be in the same GCS (Monaghan et al., 2013).

No study has yet given attention to the impacts of incorrect coordinate systems on the transport of an atmospheric tracer. Sensitivity experiments were conducted to quantify the impact of geographic coordinate systems on the atmospheric mixing ratios of methane ($CH_4$) emitted from the Marcellus shale gas production activities in Pennsylvania. Using a chemistry module
to transport passive tracers in the atmosphere, WRF simulates the $CH_4$ mixing ratios in the atmosphere.

The objectives of this study are the following:

1. Quantify the impact of projecting the model input data with different coordinate systems on meteorological variables and simulated atmospheric mixing ratios of a passive tracer.

2. Generate a tool that can automatically convert WRF output to GIS layers and vice-versa.



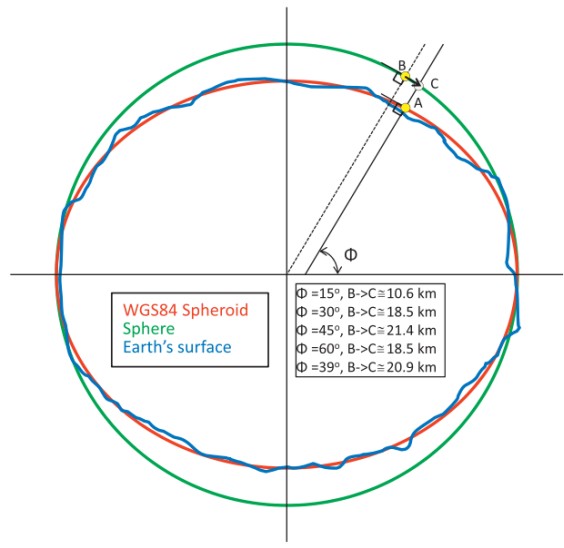

**Figure 1.** Equivalent points comparisons when using a sphere and spheroid. Blue represents the true earth shape. Green represents the speher that WRF assumes. Red shows the sperhoid WGS84 GCS. Point A represents data projected on a spheroid system. When that same point A is represented on a sphere like in an atomspherical model , its location gets incorrectly shifted to point B. Point C is the true location of point A when correctly projected in the spherical coordinate system.

**Figure 2.** Errors introduced by the different geographic coordinate system are a function of latitude. The maximum error of about 21 km is found at 45 degree latitude. The three shaded areas indicate the latitudinal extents of the three nested WRF domains used in this study.

## 1.1 Study area

The atmospheric simulations were performed using three nested domains of decreasing area and increasing spatial resolutions. As suggested by Monaghan et al. (2013), we defined several criteria to select a region where errors introduced by GCS are more likely to affect our simulation results. First, the region should have larger elevation gradients. Second, it should contain diverse land use patterns such as forest, urban, and wetland. Third, the simulation period requires convective conditions such as summer time since both the topography and the land cover play a larger effect on the simulations. Finally, a comparatively small domain should provide a focused study region because a larger domain would ignore the small variations.

The WRF model grid configuration used in this research contains three nested grids: 9×9 km for domain 1, 3×3 km for domain 2, and 1×1 km for domain 3 (Figure 3). Each 9×9 and 3×3 km grid have a mesh of 202×202 grid points. The 1×1 km grid has a mesh of 240×183 grid points.



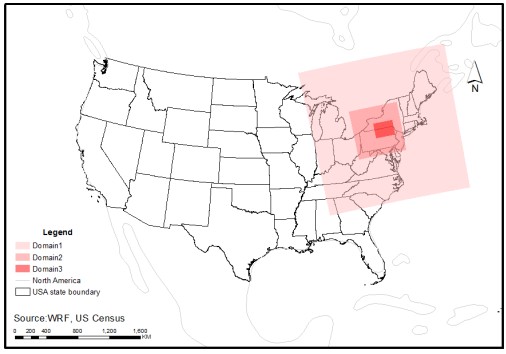

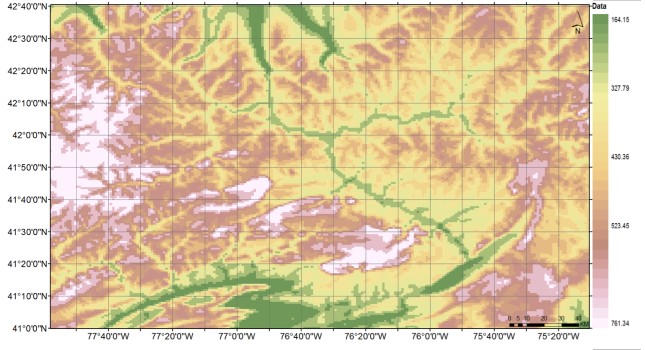

**Figure 3.** Map of study area shows three nested domains of WRF. The inner domain is located in the north eastern Pennsylvania and extents into Southeast New York.

**Figure 4.** Elevation of the WRF inner domain shows the elevation ranges from 108 m to 761 m.

The 9×9 km grid (domain 1) contains the mid-Atlantic region, the entire northeastern United States east of Indiana, parts of Canada, and a large area of the northern Atlantic Ocean. The 3×3 km (domain 2) grid contains the entire state of Pennsylvania and southern New York. The 1×1 km (domain 3) grid contains northeaster Pennsylvania and southeastern New York. One-way nesting is used so that information from the coarse domain translates to the fine domain but no information from the fine domain translates to the coarse domain (Barkley et al., 2016). The elevation of the domain 3 ranges between 108 and 706 meters above sea level.

The analysis of model results focuses on domain 3. This region was primarily chosen because there has been an increase of activity in natural gas fracking since 2008, which is expected to result in significant releases of fugitive greenhouse gas emissions, in particular $CH_4$ (Barkley et al., 2016).

## 2  Data

| Variables | DEFAULT scenario | HR scenario | HR_RESHIFT scenario |
|---|---|---|---|
| Topography | USGS | SRTM | SRTM |
| land use | USGS | NLCD | NLCD |
| Coriolis | E & F parameters | E & F parameters | E & F parameters |
| Leaf Area Index | MODIS climatology | 8-day MODIS | 8-day MODIS |
| Albedo | MODIS climatology | 8-day MODIS | 8-day MODIS |
| $CH_4$ Emissions | Barkley et al. (2016) | Barkley et al. (2016) | Barkley et al. (2016) |

**Table 1.** The table showing the input data sources for each of the three scenarios(DEFAULT, HR and HR_RESHIFT).





Table 1 shows the input data sources for each of the three scenarios. The variables include topography, land use, Coriolis, Leaf Area Index, Albedo and CH$_4$ emissions.

## 2.1 Digital Elevation Data

Two types of elevation data are included in the experiments. The WRF DEFAULT elevation data are derived from the U.S.
Geological survey (USGS) Global 30 arc seconds (roughly 900 m) elevation dataset topography, and are used in the DEFAULT case (Gesch and Greenlee, 1996). The HR and HR_SHIFT cases use higher resolution data from the NASA Shuttle Radar Topographic Mission (SRTM) (Farr et al., 2007). The data consist of a 90 m resolution Digital Elevation Model (DEM) for over 80% of the world. The data are projected in a geographic (lat/long) projection with the WGS84 GCS.

## 2.2 Land Cover Data

The DEFAULT scenario uses the 24 types of land use categories derived from satellite data and are in the WGS84 GCS and are used in the DEFAULT case. The HR and HR_SHIFT cases use the latest landcover products available for Northern America. The 2011 USGS National Land Cover Database (NLCD), covers the continental United States including the state of Alaska and are derived from Landsat satellite imagery with a 30 m spatial resolution. Furthermore, the product is modified from the Anderson Land Cover Classification System and is divided into 20 different land cover types. It has a NAD 1983 GCS and is
projected using an Albers conic equal area projection (Homer et al., 2007).

Due to the extent of the NLCD data set, the 2010 North American Land Cover (NALC)[1] is used for the areas of the domain that includes Canada. The NALC product is constructed from observations acquired by the Moderate Resolution Imaging Spectroradiometer (MODIS) at a 250 m spatial resolution. This product is produced by Canada, the US, and Mexico and is represented based on three hierarchical levels using the Food and Agriculture Organization (FOA) Land Classification System.
NALC is based on a sphere GCS with a radius of 6,370,977 m and has a Lambert Azimuthal Equal Area projection (Latifovic et al., 2012).

## 2.3 Leaf Area Index

The Leaf Area Index (LAI) variable estimates the tree canopy area relative to a unit of ground area (Watson, 1947). Two types of LAI data are used in this experiment. WRF DEFAULT LAI is based on a climatology derived from MODIS is used in
the DEFAULT scenario. LAI in HR was obtained from 8-day-averaged data from MODIS. The level-4 MODIS global LAI product composites data every 8 days at 1 km resolution on a sinusoidal grid (NASA LP DAAC, 2015a). The product we used is MCD15A2 for May 2015, which combines the MODIS data from Terra and Aqua satellites.

---

[1]2010 North American Land Cover at 250 m spatial resolution. Produced by Natural Resources Canada/ The Canada Centre for Mapping and Earth Observation (NRCan/CCMEO), United States Geological Survey (USGS); Insituto Nacional de Estadística y Geografía (INEGI), Comisión Nacional para el Conocimiento y Uso de la Biodiversidad (CONABIO) and Comisión Nacional Forestal (CONAFOR)





## 2.4 Albedo

Surface albedo is one of the key radiation parameters required for modeling of the earth's energy budget. In the DEFAULT scenario, albedo use the values from the MODIS modified by National Oceanic and Atmospheric Administration (NOAH) according to green fraction (Chen and Dudhia, 2001).

The HR and HR_RESHIFT cases use the satellite observations that are retrieved from MODIS to produce high-resolution and domain specific albedo input. A 16-Day L3 Global 500 m MCD43A3 product is used for May 2015. The product relies on multiday, clear-sky, atmospherically-corrected surface reflectances to establish the surface anisotropy and provide albedo measures at a 500m resolution (NASA LP DAAC, 2015b).

## 2.5 CH$_4$ emissions

CH$_4$ emission sources include unconventional wells and conventional wells. Both the location and amount of production rate are provided from the Pennsylvania Department of Environmental Protection (PADEP) Oil and Gas Reporting website, New York Department of Environmental Conservation, and the West Virginia Department of Environmental Protection (WVDEP). The emission was calculated by multiplying the production with the emission factors. Omara et al. (2016) indicates that the emission rate for conventional wells is 11% and unconventional well is 0.13% of the well production. The CH$_4$ emission files

were converted as input files for the WRF model (Barkley et al., 2016).

## 2.6 Weather Stations

The weather observations are the standard measurements of wind, temperature and moisture fields from World Meteorological Organization (WMO) surface stations at hourly intervals and radio sondes at 12-houly intervals. The objective analysis program OBSGRID is used for quality control to remove erroneous data (Deng et al., 2009; Rogers et al., 2013). There are 8 stations

located in the inner domain. Temperature data during the experiement time from each tower are collected to validate the model simulation results.

## 3   Methodology

The WRF model (Skamarock and Klemp (2008)) version 3.6.1 is used to generate the numerical weather simulations in this research. It is one of the most widely distributed and used mesoscale Numerical Weather Prediction (NWP) models

in existence. It has well-tested algorithms for meteorological data assimilation and meteorological researches and forecast purposes. The WRF model carries a complete suite of atmospheric physical processes that interact with the model's dynamics and thermodynamics core (Barkley et al., 2016).

   The model physics of the WRF configuration in this research includes the use of the following settings (Barkley et al., 2016). First, the double-moment scheme is used for cloud microphysical processes (Thompson et al., 2004). Second, the Kain-Fritsch

scheme is used for cumulus parameterization on the 9-km grid (Kain and Fritsch, 1990; Kain, 2004). Third, the Rapid Radiative





Transfer Method is applied to general circulation models (GCMs) (Mlawer et al., 1997; Iacono et al., 2008). Next, the level 2.5 TKE-predicting MYNN planetary boundary layer (PBL) scheme (Nakanishi and Niino, 2006), and the Noah 4-layer land-surface model (LSM) that predicts soil temperature and moisture in addition to sensible and latent heat fluxes between the land surface and atmosphere are included (Chen and Dudhia, 2001; Tewari et al., 2004; Barkley et al., 2016).

5      The WRF model enables the chemical transport option within the model allowing for the projection of $CH_4$ concentrations throughout the domain. Surface $CH_4$ emissions used as input for the model come from the $CH_4$ emissions inventory. WRF is able to simulate the $CH_4$ transport in the atmosphere.

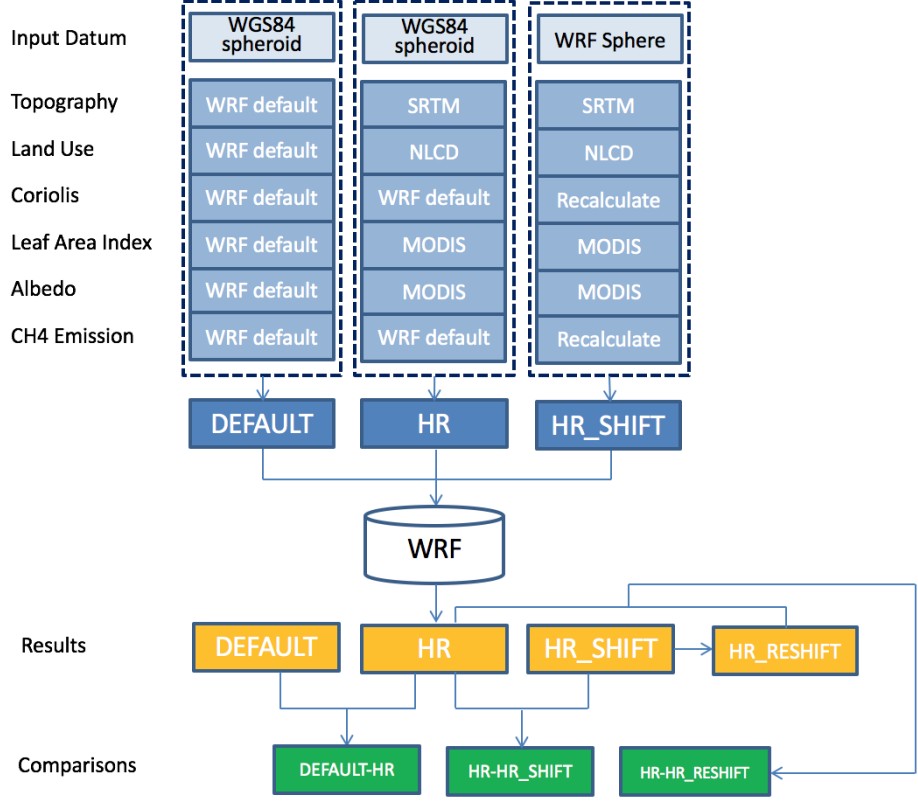

**Figure 5.** Workflow of the study showing the three scenarios: DEFAULT, HR, HR_SHIFT.

WRF simulations are performed for a 25-hour time period from 07h00 on 14[th] May 2015 until 07h00 am 15[th] May 2015 Eastern Standard Time (EST) over the three nested domains described in Section 1.1. A series of numerical weather simulations were performed using the following input datasets:

1. DEFAULT scenario: DEFAULT WRF topography, land use data, Coriolis E and F, leaf area index, albedo and $CH_4$ source emissions which are all in WGS84 GCS. The datasets are used as input without applying any transformations into WRF.





2. HR scenario: High resolution terrain and land cover data which are all in WGS84 GCS. The datasets are used as input without applying any transformations into WRF.

3. HR_SHIFT scenario: High resolution terrain, land cover data, Coriolis, leaf area index and albedo data which are first reprojected onto a spherical coordinate system using the transformation function (Hedgley Jr, 1976).

This is a summary of the comparison that are performed to assess the hypothesis.

1. DEFAULT is compared to HR to investigate the impacts on the high resolution input data on model results.

2. HR is compared to HR_SHIFT to investigate the impacts of geographic coordinate system change on model results.

3. HR_RESHIFT is originally the model output from HR_SHIFT simulation. Then, the output is shifted back to WGS84. HR_RESHIFT is compared to HR. These two outputs are in the same geographic coordinate system. The model output
comparison, such as temperature, wind speed, wind direction and CH$_4$ concentration, leads to sensitive understanding of how latitude-dependent variables affect the model simulation.

The input data include elevation, land use, Coriolis E and F components, LAI, albedo and maps of CH$_4$ sources. The CH$_4$ sources include conventional wells and unconventional wells. According to Refslund et al. (2013), using high resolution green fraction data does not significantly impact the performance of the weather model simulation. Thus, we did not replace green
fractions in this experiment.

The first simulation (DEFAULT scenario) uses the WRF DEFAULT setting: U.S. Geological survey (USGS) Global 30 arc second elevation dataset topography (GTOPO30; Gesch and Greenlee 1996), 24 types land use data, Coriolis parameters E and F, original WRF leaf area index and albedo. In addition to above variables, the experiment takes CH$_4$ emissions from unconventional and conventional wells as an input to the WRF simulation.

The second simulation, HR, uses higher resolution datasets for terrain, land cover, LAI and albedo. The terrain elevation data are derived from the NASA SRTM Digital Elevation Model (DEM) product at a 90 m resolution. The NALC and NLCD are used for the land cover data. LAI and albedo are retrieved from MODIS in May 2015. All these data are replaced for all of the three WRF domains. A common approach to re-sample land cover categories to a cell is based on the highest number of pixels that represent a class. Then the highest class occurrence is used to assign the land cover type of the cell. For example if cell
A is made up of three different land cover types: 1) Open water 38%, 2) Deciduous Forest 32%, and 3) Evergreen Forest 30% then the final class for cell A would be open water. However, in this work, a hierarchical classification scheme is used to define the land cover type. First, we determine the most common class of land cover types presents inside the cell and create a count order based on the values inside that class. A class corresponds to multiple land cover types. For example, the class "Forest" includes the types "Deciduous Forest" and "Evergreen Forest". We assign the prevalent class, such as Forest, to the given pixel.
Second, the grid cell is attributed a land covert type by selecting the type with largest values that are present within a class. For example, if the same cell A is made up of the three different land cover types :1)Open water 38%, 2)Deciduous Forest 32%, and 3)Evergreen Forest 30%, then the final class for cell A would be "Deciduous Forest" because the class "Forest" is most common class (62%) within this cell and "Deciduous Forest" has the highest percentage within the "Forest" class.





The third simulation, HR_SHIFT, uses the same data as the HR scenario, however, the input data are converted from WGS84 to the DEFAULT WRF sphere GCS.

Coriolis is a function of latitude and thus particularly affected by errors in GCS. Coriolis force has two components, E and F are calculated using $E = 2\Omega sin(\varphi)$ and $F = 2\Omega cos(\varphi)$ where $\Omega$ is rotation rate of the earth and $\varphi$ represents latitude. Coriolis E and F variables are recalculated in the HR_SHIFT scenario by using the reprojected latitude.

| Experiment ID | Input GCS | Output GCS |
|---|---|---|
| DEFAULT | WGS84 | WGS84 |
| HR | WGS84 | WGS84 |
| HR_SHIFT | WRF Sphere | WRF Sphere |
| HR_RESHIFT | WRF Sphere | WGS84 |

**Table 2.** shows the input and output GCS for the data used in each of the four analysis that will be performed.

Table 2 shows the input and output GCS for the topographic, land use, and CH$_4$ data used for the WRF simulations. Specifically, results discuss the output for the DEFAULT and HR, and HR and HR_RESHIFT configurations. A prototype tool is developed to automatically transfer WRF output to GIS layers.

## 4   Instructions of using R code

A series of scripts in R are provided to perform the tasks identified in the current paper. Figure 6 shows the process used to generate new input data based on additional input data and an optional coordinate transformation. This process is performed in the WRF_preprocess.R and WRF_updateNC.R scripts.

Additional scripts are provided to perform basic transformation of the input data from their original format to the lat-long WGS84 format that is used by WRF_preprocess.R to generate new model input data. For example MODIS_LAI.R is used to automatically download and reproject MODIS satellite data in a format that can be input into the WRF input file. These functions are provided to automate the process of downloading and reproject MODIS data, the same results can be achieved through several already alternatively methodologies. Essentially, the MODIS functions are wrappers around the MODIS Reprojection Tool, which is provided by NASA.

The current code assumes standard WRF input data in NetCDF format, however the script can be easily modified to accept a different input format from a model other than WRF.

### 4.1   WRF_preprocess.R

The signature for the function is as follows:

WRF.preprocess( filename.wrf,

    filename.raster,

    WRF.layer,





shift.to.sphere,

write.shapefile,

cores)

where:

- filename.wrf is the input file that contains the original WRF input files.

- filename.raster is the filename for the new data (e.g. MODIS LAI) file that is being used to replace the original WRF input.

- WRF.layer is the layer name in the WRF input file. For example HGT represents the height, F and E the coriolis latitudinal and meridional components.

- shift.to.sphere is a boolean (TRUE or FALSE) and determines if the input raster is reprojected to spherical coordinates from the original lat long WGS84.

- write.shapefile is a boolean (TRUE or FALSE) and determines if an ESRI Shapefile is generated.

- cores specifies the number of cores for parallel processing.

## 4.2   WRF_UpdateNC.R

WRF_UpdateNC.R file takes the generated Rdata files, and updates them into the original WRF input file.

The signature for the function are as follows:

load(filename.data)

put.var.ncdf(WRF.new,

WRF.layer,

WRF.data.HR)

where:

- filename.data is the Rdata generated from WRF.preprocess.

- WRF.new is an object of class ncdf.

- WRF.layer is what variable to write the data to. They could be HGT_M, LU_INDEX , F, E, LAI12M, and ALBEDO12M.

- WRF.data.HR is the values to be written.



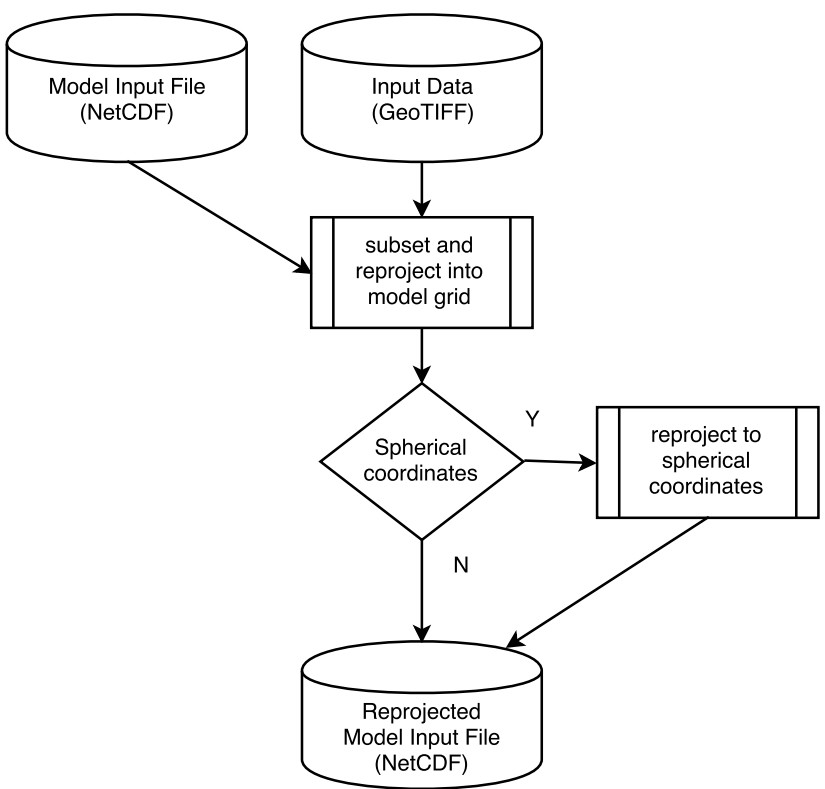

**Figure 6.** Flowchart for transforming and generating new model input data

## 5    Results

The WRF model is used to simulate the atmospheric dynamics between May $14^{th}$, 2015 07h00 and May $15^{th}$ 2015 07h00 EST. This work focuses on four output variables produced during the WRF simulation: air temperature, mean horizontal wind speed and direction, and $CH_4$ atmospheric mixing ratios. Temperature was selected because it is one of the main drivers of local and large scale weather. Additionally historical temperature data are available for comparison purposes. Near-surface temperature also corresponds to areas of higher energy which relates to turbulent motions near the surface as well as surface water exchange (evaporation). Wind speed and wind direction were selected to represent the atmospheric dynamics impacting the weather conditions at small and large scales. Finally, we selected the $CH_4$ mixing ratios to quantify the impact on greenhouse gas transport in the atmosphere.





## 5.1 DEFAULT and HR Sensitivity Study

Previous studies have investigated the weather simulation performance differences by using higher resolution data. While the comparison between DEFAULT and HR is not the central focus of this work, experiments were performed to confirm previous findings, and to quantify changes due to using higher resolution vs changes due to the different GCSs.

Figure 7, 8 and 9 compare the WRF simulations for domain 3 for temperature, wind direction and wind speed respectively. The figures show that using higher resolution data does not significantly alter the results obtained using the DEFAULT WRF input.

## 5.2 HR and HR_RESHIFT Sensitivity Study

This section analyzes the main hypothesis of the article, namely investigating the effect of using a different geographic
coordinate system on the simulations of temperature, wind speed, wind direction and $CH_4$ mixing ratio.

### 5.2.1 Results for Temperature

The effect of using a different coordinate system on the simulations of temperature is performed by comparing observations between the un-shifted (HR) and shifted (HR_SHIFT) scenarios. Figure 10 shows the difference obtained for May 14th, 2015 at 15h00. This particular time and day were chosen because it is one of the hottest times of the day, when temperature are
expected to vary the most. Letters A - H represent the eight weather observation stations located inside the selected domain and are used for validation purposes.

The temperature difference ranges from -5.6 °C represented by light blue colors to 6 °C shown with orange/red colors. When comparing both HR and HR_RESHIFT, the most striking spatial pattern is the systematic cooling around the finger lakes (roughly bound by points A, B and H). There are several additional areas of increased positive and negative temperature
around the perimeter of the image, where most extremes are observed. However, these are likely to be artifacts introduced by the WRF computations where the nested grids meet.

Statistical tests were performed using the observed weather data (stations A-H), and both scenarios (HR and HR_RESHIT) have a 0.91 root mean square error. While this suggests that there are only small temperature variations when using a different GCS, it should be noted that this test was performed only at eight stations throughout the domain where ground data were
available. Unfortunately, several of these stations lie close at the edge of the domain, where WRF simulation results are most unreliable. Therefore, the spatial cooling observed around the lakes is the most important results obtained entirely due to the change in GCS.

Both domain 2 and domain 3 show a systematic temperature increase in the HR_RESHIFT scenario when compared to HR (Figure 11 and Figure 12). The height is represented on the vertical axis while the temperature difference is on the
horizontal axis. The variability and mean temperature differences are larger near the surface and below 1 km altitude. This height corresponds approximately to the average boundary layer height where the impact of the surface on the atmospheric





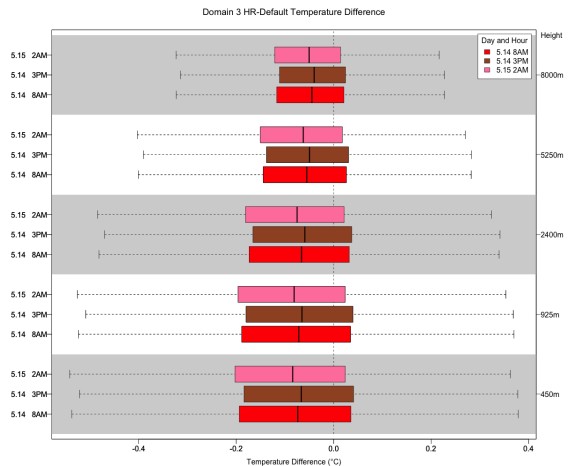

**Figure 7.** Temperature differences between HR and DEFAULT in the domain 3.

**Figure 8.** Wind direction differences between HR and DEFAULT in the domain 3.

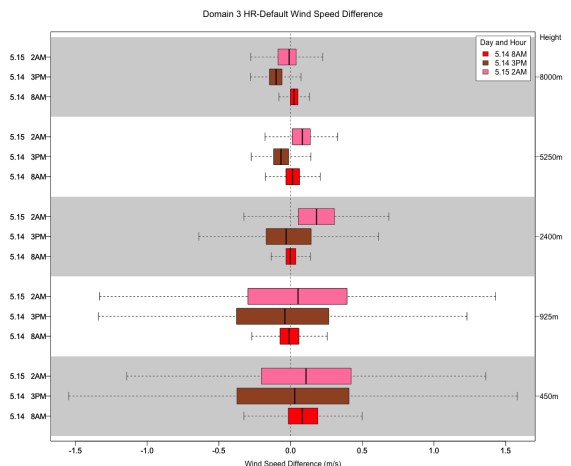

**Figure 9.** Wind speed differences between HR and DEFAULT in the domain 3.





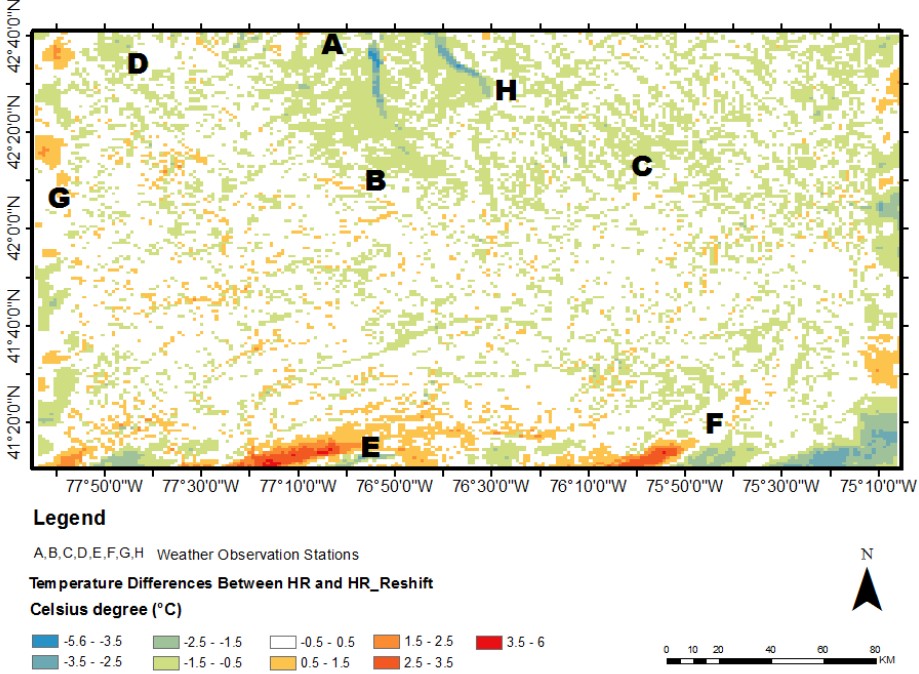

**Figure 10.** Temperature difference between HR and HR_RESHIFT on May 14th 15h00, 2015 showed that there is no significant spatial pattern.

dynamics is maximum. The variability in the mid Troposphere decreases significantly, revealing a lower impact of the GCS on the higher altitude model results.

### 5.2.2 Results for Wind speed

Figure 13 shows the wind speed difference for May 14th, 2015 at 11h00, which ranges from -5.11 to 3.5 m s$^{-1}$ between HR and HR_RESHIFT. A wave pattern is found during the 25 hours simulation. National Weather Service website operated by the National Oceanic and Atmospheric Administration (NOAA) showed a high pressure system present in the northeastern part of the domain 3. The changes in GCS seem to better represent the atmospheric dynamics of the day centered around the high pressure system. The contour of the high pressure has high correlation with the wave pattern of wind speed (Figure 14). The wind speed differences between HR and HR_RESHIFT indicate that the change in GCS affects the results.

### 5.2.3 Results for Wind direction

Figure 19 and Figure 20 show results for wind directions, and point out that, as for the previous cases, the most differences are found closer to the surface. As explained earlier, changes in GCS affect the interaction in the lower layers of the troposphere the most .



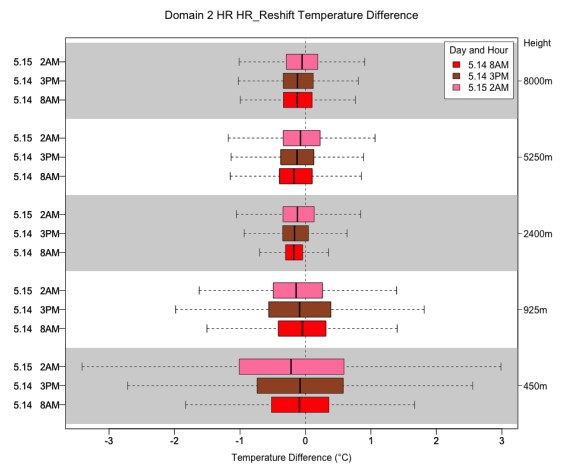

**Figure 11.** Temperature differences between HR and HR_RESHIFT in domain 2.

**Figure 12.** Temperature differences between HR and HR_RESHIFT in domain 3.

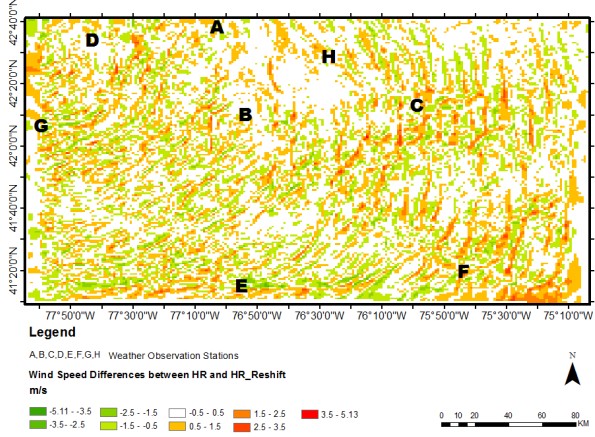

**Figure 13.** Wind speed difference between HR and HR_RESHIFT on May 14$^{th}$ 15h00, 2015 showed a wave pattern.

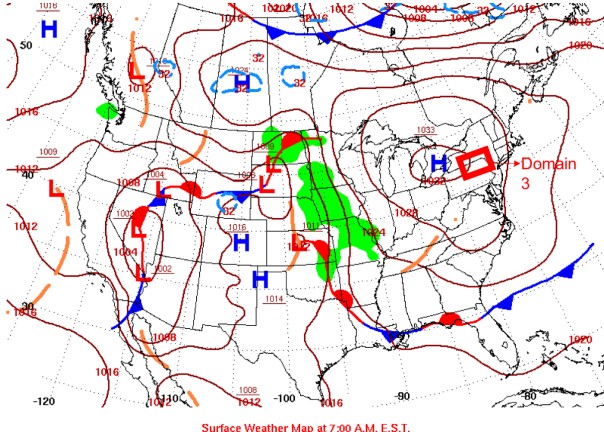

**Figure 14.** High pressure system was found in the northwestern corner of the inner domain (NOAA). It explained the wave pattern in the wind speed.





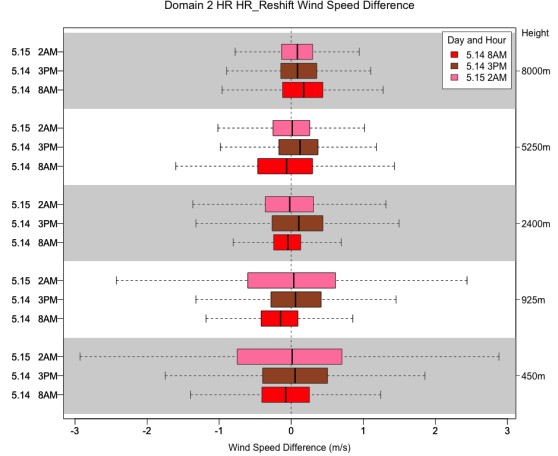

**Figure 15.** Wind speed differences between HR and HR_RESHIFT in the domain 2.

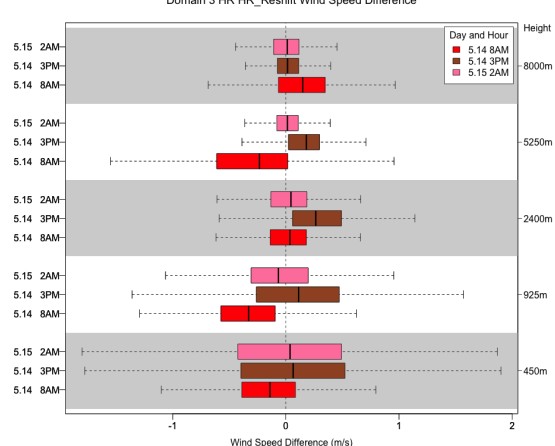

**Figure 16.** Wind speed differences between HR and HR_RESHIFT in the domain 3.

In the northeastern corner of the inner domain, there is a strip-like pattern, with large local wind changes between positive and between positive and negative North-East and North-West, and South-East and South-West. In this region the Appalachian mountains create a complex terrain with series of valley and ridges. The GCS changes the spatial distribution of the terrain elevation, leading to these very large changes in wind direction The strong vertical gradients observed in the figure suggest there there is also a combination of influences from both the surface parameters (primarily elevation and land cover), and the Coriolis components. Despite observed changes throughout the vertical column, the near-surface variability is significantly larger than the mid-Tropospheric variances as observed for temperature and wind speed.

### 5.2.4 Results for CH$_4$ Atmospheric Mixing Ratios

WRF was used to simulate CH$_4$ atmospheric mixing ratios originated from leaks from unconventional and conventional natural gas production activities respectively during the 25 hours simulation. CH$_4$ mixing ratio is a unique tracer to study atmospheric dynamics and well suited for this experiment because domain 3 includes the northeastern Pennsylvania which, since 2008, has became one of the most important fracking area in the United States since 2008. With the development of fracking, the CH$_4$ leaks became a concern because CH$_4$ has a Global Warming Potential (GWP) between 28 to 36 during 100 years. It means that the comparative impact of CH$_4$ on climate change is 28 to 36 times greater than CO$_2$ over a 100-year period (US EPA, 2015).

CH$_4$ mixing ratios are computed differently than temperature, wind speed and wind direction. Temperature, wind speed and wind directions are computed using global atmospheric input data, which is an internal variable of the WRF model physics. On the other hand, CH$_4$ mixing ratios are computed solely on the CH$_4$ emissions created using multiple datasets. Thus, CH$_4$ mixing ratios were selected to investigate the impact of differences in GCS on the simulation accuracy aggregated over time, as CH$_4$ accumulates differences along its trajectories in the atmosphere. Overall, we expect a strong sensitivity to transport



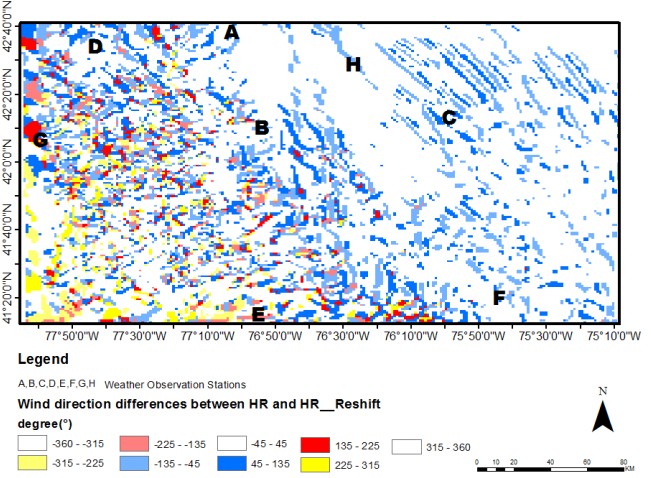

**Figure 17.** Wind direction difference between HR and HR_RESHIFT on May 14th 15h00, 2015 showing a strip pattern in the right top corner where it is a valley region. The pattern indicates that WRF model reacts differently on a small area weather simulation when the GCS changes.

**Figure 18.** Domain 3 topography map. The elevation ranges from 164 m to 761 m above the sea level.

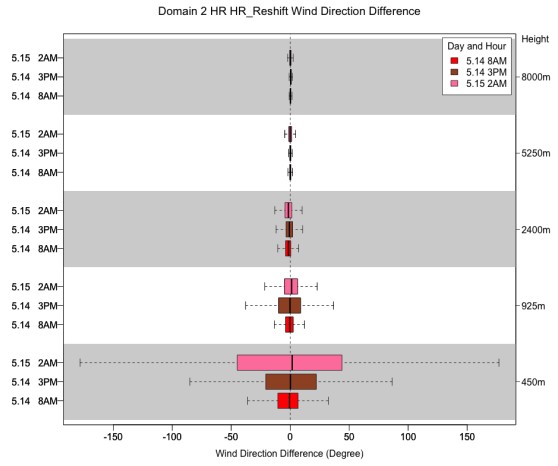

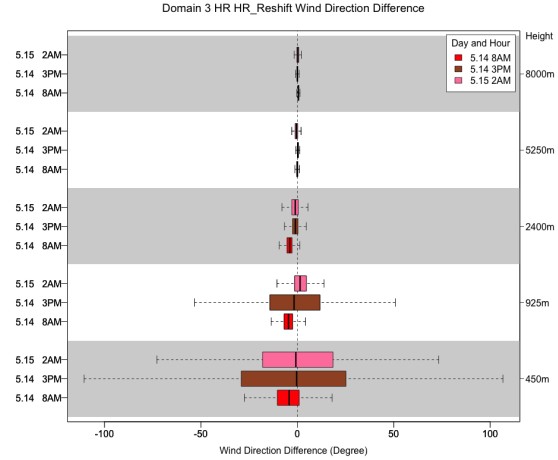

**Figure 19.** Wind direction differences between HR and HR_RESHIFT in the domain 2.

**Figure 20.** Wind direction differences between HR and HR_RESHIFT in the domain 3.




differences revealed by the long range transport of CH$_4$ emitted at the surface. Figure 21 and Figure 22 show the mean of CH$_4$ mixing ratios differences between HR and HR_RESHIFT for conventional and unconventional wells as a function of time. The figure show two radar plots, where the times have been arranged as on a clock. The left image indicates the results for AM and the right image for PM. When the shade area is larger than 0, CH$_4$ mixing ratios in HR is larger than it in HR_RESHIFT, and

5   vise versa.

For conventional wells (Figure 21), the differences are often close to 0, with night time increases (21h00 to 04h00). For the unconventional wells (Figure 22), CH$_4$ mixing ratio in HR is also smaller during night time (21h00 to 08h00), but much more so (as much as 1 PPB smaller). The reason for this change is because during night time the mixing within the boundary layer is smaller (more stable atmosphere) and therefore the magnitude of the concentration of CH$_4$ are higher. Because of the higher

10   concentrations, the impact of the change in GCS is bigger. Furthermore, the explanation as to why conventional wells have a smaller variation that unconventional wells is because most of them are are located farther away from the tower network, and thus their emission contribution on the simulation is smaller because distributed over a wider area. These results show a significant change in the CH$_4$ mixing ratio when using the different GCS.

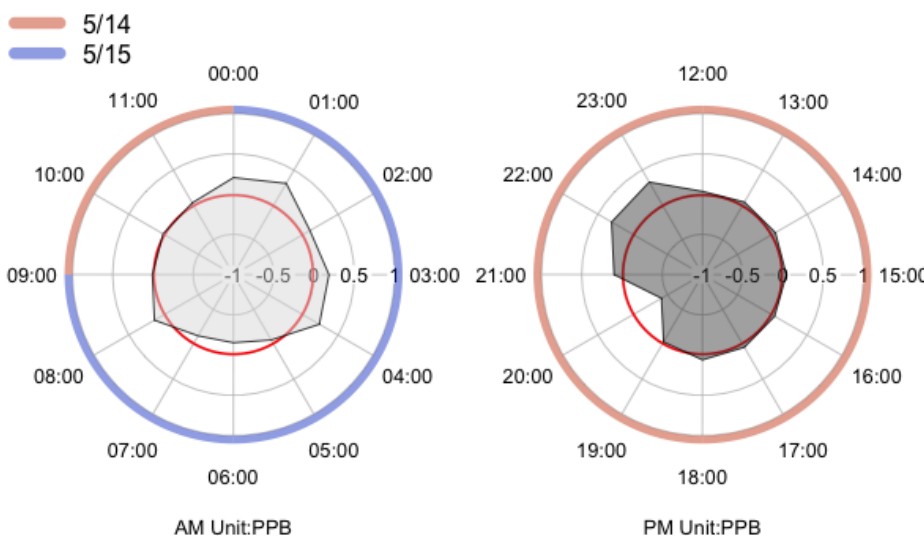

**Figure 21.** CH$_4$ mixing ratios difference between HR and HR_RESHIFT in Domain 3 for conventional wells. Left figure shows the morning time differences including 01h00 to 12h00 in May 14[th] and May 15[th]. The right figure shows the afternoon until midnight differences between 13h00 to 24h00 in May 14[th].





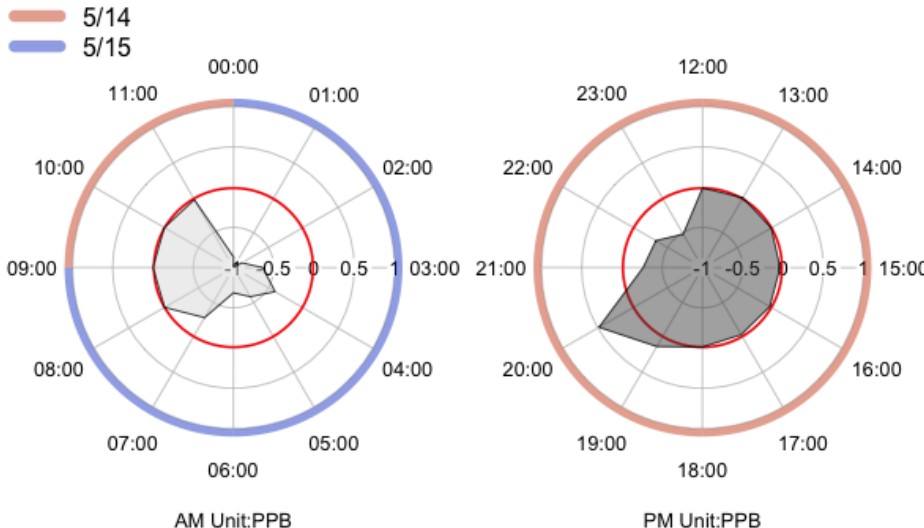

**Figure 22.** $CH_4$ mixing ratios difference between HR and HR_RESHIFT in Domain 3 for Unconventional wells. Left figure shows the morning time differences including 01h00 to 12h00 in May 14[th] and May 15[th]. The right figure shows the afternoon until midnight differences between 13h00 to 24h00 in May 14[th].

## 6 Conclusions

This paper discusses the impact of different geographic coordinate system on weather numerical model simulations. The main hypothesis is that the error introduced by not taking into account the GCS of the input data, which might result in latitudinal errors of up to 21 km in the mid latitudes, can cause significant changes in the output of the model.

A sensitivity study was performed using the WRF numerical model, with input data at different resolutions and different GCSs. Four different output parameters were investigated, namely temperature, wind speed, wind direction and $CH_4$ mixing ratios.

Results show that changes are introduced by using different GCSs for the input data. The observed differences were caused by 1) topography shift including elevation, land use, albedo, LAI differences; and 2) latitude-dependent physics, such as the Coriolis force and the incoming solar radiation.

A systematic temperature increase was observed in all of the three domains used in this study. A spatial pattern showing significant cooling was observed near two lakes included in the inner domain.

Similarly, wind speed and direction show spatial changes that can be traced back to the use of a different land cover and elevation. Wind speed, wind direction and temperature indicate more variations within the planetary boundary layer where the



interaction between the surface and the atmosphere is greatest. It is expected that changes at the surface will introduce most significant changes closer to the surface.

it is shown that, without exceptions, the GCS of the input data affects model results. Sometimes these changes are large and have a clear spatial patterns, whereas other times are small and negligible. It is concluded that while some of of these

5 errors might be small, they nevertheless introduce an additional bias in the model output. Especially for very high resolution simulation, these errors are compounded and can lead to significant errors.

While it is best to properly project all data in the correct representation used by the model, which in the case of WRF is a spherical GCS, it is most important to keep the GCS and projection among the input layers consistent. In fact, if all layers are in the same GCS, errors in mapping onto the surface of the earth are consistent across the datasets and the effect of using the

10 wrong GCS are minimized. On the other hand, mixing GCSs in the input data leads to larger errors.

## 7  Code availability

WRF processing code is available at https://github.com/yannicao/wrf_reprojection.

## 8  Acknowledgments

This research was partially supported by the Department of Energy (DE-FE0013590) and by the Office of Naval Research

(ONR) award #N00014-16-1-2543 (PSU #171570)





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
