# Peer review of "Influence of Geographic Coordinate System on Weather Simulations of Atmospheric Emissions"

_Geoscientific Model Development, 2016_

## Short Comment (SC1) · 12 Dec 2016

Dear authors,

in my role as Executive editor of GMD, I would like to bring to your attention our Editorial version 1.1:

http://www.geosci-model-dev.net/8/3487/2015/gmd-8-3487-2015.html

This highlights some requirements of papers published in GMD, which is also available on the GMD website in the 'Manuscript Types' section:

http://www.geoscientific-model-development.net/submission/manuscript_types.html

In particular, please note that for your paper, the following requirements have not been met in the Discussions paper:

- "If the model development relates to a single model then the model name and the version number must be included in the title of the paper. If the main intention of an article is to make a general (i.e. model independent) statement about the usefulness of a new development, but the usefulness is shown with the help of one specific model, the model name and version number must be stated in the title. The title could have a form such as, "Title outlining amazing generic advance: a case study with Model XXX (version Y)"."

Therefore, please add something like " a case study using WRF (version Y.Z)" to the title.

Furthermore, I'd like to add, that in my opinion the title of your paper is hardly understandable. Therefore I suggest to reformulate it. What about something like: "How does the usage of the Geographic Coordinate System influence weather dependent trace gas emissions? A case study using WRF (version Y.Z)" or "Influence of the usage of the Geographic Coordinate System on weather dependent emissions: A case study using WRF (version Y.Z)" ?

Yours,

Astrid Kerkweg

---

## Author Comment (AC1) · 10 Feb 2017

Dear Dr. Kerkweg,

Thank you very much for your comments. This research is a model independent study. We changed some variables input in WRF. But we can conduct this research using other WRF updates or even other atmospheric numerical models. Therefore, we consider it will be misleading to have WRF model version in the title. We will annotate the article.

Best, Yanni

---

## Referee Comment (RC1) · Anonymous Referee #1 · 15 Feb 2017

General comments: The article introduces the impact of different geographical coordinate systems to run the WRF mesoscale atmospheric model. The paper shows that an error, although small, is introduced purely through using different coordinate systems. This result is not intuitive, and its publication is commendable.

I agree with the comments made by the editor, and I would suggest to rephrase the title from its current "Influence of Geographic Coordinate System on Weather Simulations of Atmospheric Emissions" into "Analysis of errors Introduced by Geographic Coordinate Systems on Weather Numeric Prediction Modeling"

Specific comments: The paper is well explained and structured, and its publication is commendable. I feel that all sections are appropriate and needed, and the text is appropriate. However, some of the figures require improvements.

Virtually all figures have too small fonts. The figures have to be regenerated, and the fonts must be kept at a minimum of size 9. It is also not needed to add longitudes and latitudes at such a fine resolution, and it is best to reduce them for improved clarity. I suggest to either reproduce the figures using larger fonts, or perhaps increase their sizes in the paper. There are a lot of whitespaces that can be omitted at the advantage of larger figures.

Figure 13 and 14 are not clear. Figure 13 shows the difference in wind speed between HR and HR_RESHIFT, and therefore the association with the high pressure (Figure 14) is not obvious. The difference in the HR and HR_RESHIFT generates a wave-like pattern, but this is not necessarily due to the high pressure, since it represents a change in coordinates. The discussion should include a detailed explanation of why and how a high pressure system, which has a wave-like pattern, generates such differences only due to the different coordinate systems. Are the authors suggesting that the different coordinate system can better understand the state of the atmosphere, and therefore correlate to the high pressure system?

Figure 17 and 18 must be top aligned. The fonts for the points A—H should be the same as for the rest of the figures. The points are too large and bold, and the axis too small.

I am intrigued as to why the errors for the nested grids tend to be symmetric around 0. I suspect that it is because WRF tries to conserve energy across the nested grids, and consequently the errors for each grid must sum up to zero.

I have some problems understanding figure 22 and 23. Why are the data on the left figure for both 5/14 and 5/15?

The graph in Figure 1 should include a reference to Monaghan et al in the caption.

---

## Referee Comment (RC2) · Anonymous Referee #2 · 22 Feb 2017

General comments: This paper examines the impacts/errors of using varying coordinate systems on the model output. Specifically, WRF model simulations scenarios with geographic coordinate system (GCS) and role the role and importance of reprojecting GIS layers. Since significant errors could be introduced using different project systems, it would be very useful to quantify such impact. Therefore, this study is very needed and relevant to GIScience and Earth Science fields. The paper is clear and well written.

Specific comments:

Some improvements are suggested as below.

1. The development of a tool for WRF output and GIS layers is considered as one of the study goals. Therefore, the authors are encouraged to have a section/paragraph to indicate the motivation and the state-of-art works on developing GIS tools for processing,

mapping and visualizing atmospheric model output.

2. The section Study area is numbered 1.1, and there is no 1.2, etc. Therefore, you can directly change it to section 2, or merge it to next Data section as Study area and Data.

3. The title of the current Section 4 "Instructions of using R code" is too technical. Perhaps change to something more scientific, e.g., "WRF model input and output processing"?

4. Page 9, Line 18, it would be useful to cite or provide the link for NASA tool.

5. Current Section 4.1 and 4.2 focusing on introducing the program function and associated parameters, do not contain too much useful information for the audiences. The authors could remove or condense these two sections, put detailed codes as appendix instead, and discuss the (re-)projection tool and development in more details to match your study goal.

6. The temperature difference between HR and HR_RESHIT in Figure 10 is quite interesting. It seems like the impact is more significant in the areas close to the border. At the same time, the simulation results by atmospheric modelling could also be less accurate at these areas. Is the any linkage?

7. Right now, Figure 4 about elevation ranges is difficult to interpret for general readers. Instead, a reference map (e.g., google map?) showing the major geographic features (e.g., Finger lakes) and landmarks for the domain 3 could be more helpful for us to understand the results in Figure 10, 13 etc.?

8. Is it possible to use any ground truth data (e.g., from monitoring stations or remote sensing) to compare them with the simulation results (for one more parameter(s), e.g., temperature)? Correspondingly, more meaningful conclusion can be drawn.

Typos: 1. Page 6, line 3 NOAH -> NOAA 2. Page 9, line 16, reproject MODIS -> reprojecting MODIS 3. Page 20, line 3, it is -> It is

---

## Author Response (AR1)

Responses to reviewers 1 and 2

**General Comments** We thank the editor and the anonymous reviewers for the useful comments that improved the original manuscript in many ways. We provide a new manuscript that is a significant improvement over the original submission. We addressed all comments provided, some of which requested sections to be resized or omitted, and others to be expanded or added.

This is a list of major changes that are included in the present resubmission:

- The title has changed from "Influence of Geographic Coordinate System on Weather Simulations of Atmospheric Emissions" to "Analysis of Errors Introduced by Geographic Coordinate Systems on Weather Numeric Prediction Modeling"

- New figures were generated according to the comments received. All the figures were changed to larger font. A lot of white spaces were omitted at the advantage of larger figures.

- We remove the figure and content about the high pressure system. We agree with the reviewer that more evidence is needed to associate the high pressure system and the wind speed wave patterns.

- We reorganized sections. Study Area was changed to section 2. The title of section 5 "Instructions of using R code" was modified to "WRF model input and output processing". An appendix section was also added at the end of the paper to introduce the details of the R code used to perform all the tasks described.

**First Review**

Our responses are reported below, along with the unabridged comments of the **first** reviewer.

> [Reviewer's Comment] General comments: The article introduces the impact of different geographical coordinate systems to run the WRF mesoscale atmospheric model. The paper shows that an error, although small, is introduced purely through using different coordinate systems. This result is not intuitive, and its publication is commendable.

We thank the reviewer for the encouraging comments. We have revised the manuscript according to all comments we have received, and we hope that this new manuscript will be found much improved compared to the original submission and acceptable for publication.

> I would suggest to rephrase the title from its current "Influence of Geographic Coordinate System on Weather Simulations of Atmospheric Emissions" into "Analysis of errors Introduced by Geographic Coordinate Systems on Weather Numeric Prediction Modeling"

We have changed the title according to this comment.

> Virtually all figures have too small fonts. The figures have to be regenerated, and the fonts must be kept at a minimum of size 9. It is also not needed to add longitudes and latitudes at such a fine resolution, and it is best to reduce them for improved clarity. I suggest to either reproduce the figures using larger fonts, or perhaps increase their sizes in the paper. There are a lot of whitespaces that can be omitted at the advantage of larger figures.

We have regenerated almost all the figures. Figure 7, 8, 9, 11, 12, 14, 15, 18, and 19 are regenerated with larger size labels. Figure 10,13,16, and 17 are resized to show clearer figures.

> Figure 13 and 14 are not clear. Figure 13 shows the difference in wind speed between HR and HR_RESHIFT, and therefore the association with the high pressure (Figure 14) is not obvious.

We agree with the reviewer that more scientific evidence is needed to link the high pressure system and the wind speed wave pattern. Therefore, we removed the high pressure system figures and content.

> Figure 17 and 18 must be top aligned. The fonts for the points should be the same as for the rest of the figures. The points are too large and bold, and the axis too small.

All figures were lined up.

> I have some problems understanding figure 22 and 23. Why are the data on the left figure for both 5/14 and 5/15?

The WRF model run between May 14th, 2015 07h00 and May 15th, 2015 07h00 EST. The left side graph represents the morning hours, which include the data from 08h00 to 11h00, May 14th, 2015 and from 00h00 to 07h00, May 15th, 2015. Therefore, the left graph include two days.

> The graph in Figure 1 should include a reference to Monaghan et al in the caption.

We have added the suggested references. We thank the reviewer for all the comments.

**Second Review**

Our responses are reported below, along with the unabridged comments of the **second** reviewer.

> [Reviewer's Comment] General comments: This paper examines the impacts/errors of using varying coordi- nate systems on the model output. Specifically, WRF model simulations scenarios with geographic coordinate system (GCS) and role the role and importance of reprojecting GIS layers. Since significant errors could be introduced using different project systems, it would be very useful to quantify such impact. Therefore, this study is very needed and relevant to GIScience and Earth Science fields. The paper is clear and well written.

We thank the reviewer for the positive comment. We have extensively modified the original submission according to this reviewer suggestions. We have emphasized more clearly what are the main contributions of the article, and reorganized the document with more information on the introduction and appendix to clarify the issues identified by this reviewer. We hope that this new version is found to be a significant improvement over the original submission.

> The development of a tool for WRF output and GIS layers is considered as one of the study goals. Therefore, the authors are encouraged to have a section/paragraph to indicate the motivation and the state-of-art works on developing GIS tools for processing, mapping and visualizing atmospheric model output.

We added a paragraph to introduce the state-of-art works between the GIS and atmospheric sciences.

> The section Study area is numbered 1.1, and there is no 1.2, etc. Therefore, you can directly change it to section 2, or merge it to next Data section as Study area and Data. The title of the current Section 4 "Instructions on using R code" is too technical. Perhaps change to something more scientific, e.g., "WRF model input and output processing"?

We reorganized the section structures and rename the "Instructions of using R code" section to "WRF model input and output processing".

> Page 9, Line 18, it would be useful to cite or provide the link for NASA tool

We thank reviewer to suggest reference. The reference was added.

> Current Section 4.1 and 4.2 focusing on introducing the program function and associated parameters, do not contain too much useful information for the audiences. The authors could remove or condense these two sections, put detailed codes as appendix instead, and discuss the (re-)projection tool and development in more details to match your study goal.

We have added the appendix section which introduces the details of the R code.

> The temperature difference between HR and HR_RESHIT in Figure 10 is quite interesting. It seems like the impact is more significant in the areas close to the border. At the same time, the simulation results by atmospheric modelling could also be less accurate at these areas. Is the any linkage?

The border area is less accurate because the border area is influenced by a mismatch between the reprojected field and the global atmosphere input. We illustrated that in the introduction section. "a minor mismatch between the WRF model global atmosphere input and static variables will affect the simulation result."

> Right now, Figure 4 about elevation ranges is difficult to interpret for general readers. Instead, a reference map (e.g., google map?) showing the major geographic features (e.g., Finger lakes) and landmarks for the domain 3 could be more helpful for us to understand the results in Figure 10, 13 etc.?

We have changed the Figure 4 in the article to a Google basemap showing in Figure 1.

> Is it possible to use any ground truth data (e.g., from monitoring stations or remote sensing) to compare them with the simulation results (for one more parameter(s), e.g., temperature)? Correspondingly, more meaningful conclusion can be drawn.

[Figure]

Figure 1: The elevation in WRF inner domain ranges from 108 m to 761 m. The latitude ranges from 40°N to 42.67°N. The logitude ranges from -78°W to -75.17°W.

We totally agree with the reviewer that it is a good idea to use ground truth data to validate the model result. We actually did the validation and describe it in the Data section and Result section. However, the result suggests that there are only small temperature variations when using a different GCS, it should be noted that this test was performed only at eight stations throughout the domain where ground data were available.

Typos: 1. Page 6, line 3 NOAH -¿ NOAA 2. Page 9, line 16, reproject MODIS -¿ reprojecting MODIS 3. Page 20, line 3, it is -¿ It is

We have corrected all the typos.

[revised manuscript text omitted]

---

## Author Response (AR2)

Responses to editor

Dear Dr. Archibald,

Thank you very much for the thoughtful comment. We changed the R package from ncdf to ncdf4 in all of our scripts. Ncdf4 package (https://cran.r-project.org/web/packages/ncdf4/index.html) is well maintained. The revised R scripts files are updated to github (https://github.com/yannicao/wrf_reprojection).

Best,
Yanni